

# A non-extensive approach to probabilistic seismic hazard analysis

Sasan Moatghed[1], Mozhgan Khazaee[1], Nasrollah Eftekhari[2], Mohammad Mohammadi[3]

[1] Department of Civil Engineering, Faculty of Engineering, Behbahan Khatam Alanbia University of Technology, Behbahan, Iran.
[2] Faculty of Technology and Mining, Yasouj University, Choram 75761-59836, Iran.
[3] Department of Statistics, Faculty of Science, Behbahan Khatam Alanbia University of Technology, Behbahan, Iran.

*Correspondence to*: Sasan Motaghed (motaghed@bkatu.ac.ir)

**Abstract.** We modify the probabilistic seismic hazard analysis (PSHA) formulation by replacing the Gutenberg-Richter power law with the SCP non-extensive model for earthquake size distribution and call it NEPSHA. SCP claimed to better model the regional seismicity than the classical models. The proposed method (NEPSHA) is implemented in the Tehran region, and the results are compared with the classic PSHA method. The hazard curves show that NEPSHA gives a higher hazard, especially in the range of practical return periods. The uniform hazard spectra of NEPSHA provide more spectral accelerations, especially for the medium height buildings, which are the most common urban structures.

## 1 Introduction

PSHA is the most widely used approach to estimate the seismic load for use in engineering design processes. The main objective of PSHA computations is to calculate ground motions with different exceedance probabilities during a specific time interval (Anbazhagan, 2019). This information is the gateway to defining the possible scenario earthquakes and is used to develop instructions for seismic codes and standard regulations (Nath, 2022; Iervolino, 2022).

In the PSHA procedure, the average annual rate of exceeding a particular threshold value, *x*, of a ground motion intensity measure (IM), is computed as (Cornell, 1968):

$$\lambda_{IM}(x) = \sum_{i=1}^{n_{flt}} \upsilon \iint_{m\ r} G_{IM|M,R}(IM \geq x \mid m,r) f_M(m) f_{R|M}(r|m) dm dr \tag{1}$$

where, $n_{flt}$ is the number of causative faults, and *v* is the mean annual frequency of occurrence of earthquakes with magnitudes between a lower-bound threshold value, $m_{min}$, and an upper-bound threshold value, $m_{max}$. Also, *M* and *R* denote the moment magnitude and the source-to-site distance, respectively. The term $G_{IM|M,R}$ provides the probability that an IM exceeds a value of *x* given the occurrence of an earthquake of magnitude *m* at distance *r*. This term can be calculated using





ground motion prediction equations. The term $f_M$ represents the probability density function (*PDF*) of the earthquake magnitude and $f_{R|M}$ denotes *PDF* of distance r conditional on *m*.

Determining the function of $f_M$ is a challenging task in PSHA computations. This function should be calculated using the
frequency-magnitude relationship, which represents the background seismicity of the study region. Previous studies showed that the characteristics of such a relationship significantly affect the results of PSHA (Yazdani et al., 2015, Motaghed et al., 2021). Thus, there has been a continued interest in selecting the best representative expression for the frequency-magnitude relation.

Currently, the most commonly used model to reflect the frequency-magnitude distribution in the PSHA procedure is based
on the Gutenberg-Richter (GR) law (Gutenberg and Richter, 1956). This model represents a linear relationship between the logarithm of the frequency and magnitude as $\log_{10}N(m)$=a-*bM*, where *N* is the number of events with a magnitude greater than or equal to *m* in a given region and specified time period, and *a* and *b* are constants. So magnitudes obey power law distribution. $10^a$ represents the total number of earthquakes with magnitudes greater than $m_{min}$, and *b* (commonly referred to as *b*-value) is the slope of the fitted line. The *b*-value describes the specific relationship between the magnitude and the total
number of earthquakes commonly close to 1.0 in seismically-active regions. This simple linear relation can also be written in the form of $N(m)$=exp($\alpha$-$\beta M$), in which $\alpha$=$a$ln(10) and $\beta$=$b$ln(10). The *PDF* of $N(m)$ is then given by:

$$f_m(m) = \frac{\beta e^{-\beta(m-m_{min})}}{1-e^{-\beta(m_{max}-m_{min})}} \qquad ; m_{min} \leq m \leq m_{max} , \qquad (2)$$

This function is a double truncated form of GR law (Žalohar, 2018).

Contrary to the widespread use of the GR model in the PSHA studies, some researchers reported that earthquake magnitudes
do not always follow this distribution (Schwartz and Coppersmith, 1984; Youngs and Coppersmith, 1985; Wesnousky, 1994, Ishibe and Shimazaki, 2008). This is especially the case in situations where the seismic region consists of individual faults or fault segments with regular geometries (Ishibe and Shimazaki, 2008). In these situations, the GR model may not represent the seismicity over the entire magnitudes range. Also, while the GR recurrence model may well represent the distribution of small earthquake magnitudes, it underestimates the frequency of large earthquakes (Kramer, 1996; Youngs and
Coppersmith, 1985; Parsons and Geist, 2009).

To cope with these problems, some alternative models to the power-law have been developed by researchers, such as bilinear (Staudenmaier et al., 2018), quadratic law (Merz & Cornell, 1973), generalized Pareto distribution-based model, and random GR model (Dutfoy & Senfaute, 2021). Nevertheless, one of the most exciting models for earthquake recurrence has been proposed by Sotolongo-Costa and Posadas (2004), which is named SCP Model. The framework of this model has been
developed based on the Tsallis non-extensive approach (Tsallis, 1988). Generally, the non-extensive Tsallis entropy has been the focus of much attention over the last four decades. It is thought that this non-extensive formulation presents an



appropriate tool for investigating complex systems, especially in their nonequilibrium stationary states (Silva et al., 2006, Vallianatos et al., 2016, Vallianatos et al., 2018). The SCP model characterizes two profiles interacting via fragments filling the gap between them. This model has the advantage of representing the size distribution of fragments on the energy distribution of earthquakes. Also, the SCP model deduced an energy distribution function, which gives the GR law as a particular case (Telesca, 2012).

Despite its unique features, the SCP model has not yet been included directly in PSHA computations. This study aims to address this gap by providing a practical framework. To this end, the *PDF* form of the *SCP* model should be calculated and substituted in the classical PSHA integral. The details of this approach will be described in the following sections. This PSHA procedure that considers the seismicity model based on the non-extensive statistical physics is called here a non-extensive PSHA (NEPSHA). Finally, to investigate the differences between the results of NEPSHA and the classical framework of PSHA, we compare these approaches via a practical example.

## 2 On the SCP model

This section describes the non-extensive theoretical basis of the SCP model. Generally, statistical mechanics uses statistical methods to describe systems with high degrees of freedom. In this way, the randomness and chaos resulting from internal imperfections can be processed (Englman et al., 1988). To use this concept in the representation of fault rupture, the Boltzmann-Gibbs statistics can be used. The Boltzmann-Gibbs entropy, S, is given by

$$S = -k \sum_{i=1}^{W} p_i ln p_i , \tag{3}$$

where $p_i$ is the probability of the microscopic state $i$, $k$ is Boltzmann's constant, and $W$ is the total number of small-scale states (Sotolongo-Costa et al., 2000). Tsallis' statistics generalizes the Boltzmann-Gibbs statistics in what concerns the concept of entropy. It should be noted that fractioning is a paradigm of non-extensivity, since the fractured object can be regarded as a collection of divided parts with larger entropy than their union. So, if the parts or fragments in which the object is denoted by $A_i$ (*s*), its entropy, *S*, is $S(U_{Ai}) < \sum_i S(A_i)$, where *U* is the "*Union*" symbol. This inequality defines a "superextensivity" (Tsallis et al., 1998) in the system. So, it is necessary to use non-extensive statistics instead of Boltzmann-Gibbs statistics (Sotolongo-Costa et al., 2000). Such formalism has been proposed by Tsallis (Tsallis, 1988), as:

$$S_{q \neq 1} = -k_B \int p^q(\sigma) ln_q p(\sigma) d\sigma, \tag{4}$$

where $k_B$ is the Boltzmann constant; $p$ denotes the probability of finding a fragment of surface $\sigma$ (defined as a characteristic surface of the system), and $q$ is the non-extensive parameter. Accordingly, the $q$-logarithmic function is defined as:

$$ln_q p = (1-q)^{-1}(p^{1-q} - 1) \qquad p > 0, \tag{5}$$


The mechanism of triggering earthquakes is established through the combination of the irregularities of the fault planes and the distribution of fragments between them. The basic idea in the SCP model is the fact that the surfaces of the fault planes (interface) are irregular, and the fragments filling the space between them have diverse irregular shapes. Previous studies reveal that the Boltzmann-Gibbs statistics cannot account for the presence of scaling in the size distribution of fragments (Englman et al., 1988). So, violent fractioning is a nonextensive phenomenon, and a nonextensive

representation is necessary for its explanation. In the SCP model, the fragment-distribution function (EDF) emerges naturally from a non-extensive framework. So, the energy-distribution function is given by (Sotolongo-Costa and Posadas, 2004):

$$\log(N_{>m}) = logN + \left(\frac{2-q}{1-q}\right) \times log\left[1 + a_{SCP}(q-1)(2-q)^{(1-q)/(q-2)} \times 10^{2m}\right], \tag{6}$$

where $a_{SCP}$ is the constant of proportionality between released energy and fault rupture length. This expression describes the

energy distribution in all detectable ranges of magnitudes very well, unlike the empirical formula of GR (Sotolongo-Costa and Posadas, 2004).

Non-extensive models have attracted the attention of researchers in various branches of earth sciences. Some researchers have made modifications in the SCP model and tried to improve the seismicity description (Silva et al., 2006, Telesca, 2012). Due to the advantage of the non-extensive methods, researchers have tried to fit them to the reginal data, calculate the

parameters of the models and describe the seismicity (Sarlis et al., 2010, Matcharashvili et al., 2011, Valverde-Esparza et al., 2012, Vallianatos and Michas, 2020). Also, models based on Tsallis entropy have been used to determine the precursors (Eftaxias, 2010). Interestingly, these models have also been used to describe marsquakes (da Silva and Corso, 2021). Vallianatos et al. (2016, 2018) have provided two comprehensive reviews of these methods. In this way, trying to rewrite the well-known PSHA method based on the non-extensive approach can be helpful.

## 3 PSHA based on the SCP model

Equation 6 indicates the number of earthquakes in magnitude bins. In order to include this relationship in the PSHA calculations, it must be written as a distribution function, which is the core of this research and will be described in this section.

Tectonic faults produce earthquakes of various sizes (i.e., magnitudes). Regarding equation 6, the SCP model describes the

size distribution of earthquakes as

$$N_m/N = \left[1 + a_{SCP}(q-1)(2-q)^{\frac{1-q}{q-2}} \times 10^{2m}\right]^{\left(\frac{2-q}{1-q}\right)}, \tag{7}$$

If $m = m_{min}$, this equation yields:


$$N_{m_{min}}\big/_N = \left[1 + a_{SCP}(q-1)(2-q)^{(1-q)/(q-2)} \times 10^{2m_{min}}\right]^{\left(\frac{2-q}{1-q}\right)}, \tag{8}$$

Therefore, the cumulative distribution function ($CDF$) of the magnitudes of earthquakes, $F_M(m)$, larger than $m_{min}$ can be

written as:

$$F_M(m) = P(M \le m | M > m_{min}) = \frac{\text{Rate of earthquakes with } m_{min} < M \le m}{\text{Rate of earthquakes with } m_{min} < M}$$

$$= \frac{\lambda_{m_{min}} - \lambda_m}{\lambda_{m_{min}}} = 1 - \frac{\lambda_m}{\lambda_{m_{min}}}$$

$$= 1 - \frac{\left[1 + a_{SCP}(q-1)(2-q)^{\frac{1-q}{q-2}} \times 10^{2m}\right]^{\frac{2-q}{1-q}}}{\left[1 + a_{SCP}(q-1)(2-q)^{\frac{1-q}{q-2}} \times 10^{2m_{min}}\right]^{\frac{2-q}{1-q}}} \quad ; m > m_{min}, \tag{9}$$

where $\lambda_m = \frac{N_m}{time \times space}$ and $\lambda_{m_{min}} = \frac{N_{m_{min}}}{time \times space}$. In this equation, unlike the non-extensive expression of Telesca (Telesca, 2012) in which the catalog completeness magnitude is used, we include the minimum earthquake magnitude of engineering significance, $m_{min}$. We can compute the PDF of M by taking the derivative of the CDF, as

$$f_M(m) = \frac{d}{dm} F_M(m) = \frac{d}{dm}\left[1 - \frac{\left[1 + a_{SCP}(q-1)(q-2)^{\frac{1-q}{q-2}} \times 10^{2m}\right]^{\frac{2-q}{1-q}}}{\left[1 + a_{SCP}(q-1)(q-2)^{\frac{1-q}{q-2}} \times 10^{2m_{min}}\right]^{\frac{2-q}{1-q}}}\right]$$

$$= \frac{\left[1 + a_{SCP}(q-1)(2-q)^{\frac{1-q}{q-2}} \times 10^{2m}\right]^{\frac{1}{1-q}} \times a_{SCP}(2-q)^{\frac{-1}{q-2}} \times 2 \times 10^{2m} \ln 10}{\left[1 + a_{SCP}(q-1)(2-q)^{\frac{1-q}{q-2}} \times 10^{2m_{min}}\right]^{\frac{2-q}{1-q}}} \quad ; m > m_{min}, \tag{10}$$

where $f_M(m)$ denotes the $PDF$ of $M$. Note that the $PDF$ given in equation 10 relies on the $SCP$ formulation of equation 8,

which represents magnitudes without an upper limit. Earthquake magnitude essentially has an upper limit ($m_{max}$). Rewritten equation 8 with the $m_{max}$ is:

$$F_M(m) = P(M \le m | m_{min} < M < m_{max}) = \frac{\text{Rate of earthquakes with } m_{min} < M \le m}{\text{Rate of earthquakes with } m_{min} < M < m_{max}} = \frac{\lambda_{m_{min}} - \lambda_m}{\lambda_{m_{min}} - \lambda_{m_{max}}}$$

$$= \frac{\left[1 + a_{SCP}(q-1)(2-q)^{\frac{1-q}{q-2}} \times 10^{2m_{min}}\right]^{\frac{2-q}{1-q}} - \left[1 + a_{SCP}(q-1)(2-q)^{\frac{1-q}{q-2}} \times 10^{2m}\right]^{\frac{2-q}{1-q}}}{\left[1 + a_{SCP}(q-1)(2-q)^{\frac{1-q}{q-2}} \times 10^{2m_{min}}\right]^{\frac{2-q}{1-q}} - \left[1 + a_{SCP}(q-1)(2-q)^{\frac{1-q}{q-2}} \times 10^{2m_{max}}\right]^{\frac{2-q}{1-q}}}$$

$$; m_{min} < m < m_{max}, \tag{11}$$

and equation 10 becomes:




$$f_M(m) = \frac{\left[1 + a_{SCP}(q-1)(2-q)^{\frac{1-q}{q-2}} \times 10^{2m}\right]^{\frac{1}{1-q}} \times a_{SCP}(2-q)^{\frac{-1}{q-2}} \times 2 \times 10^{2m} \ln 10}{\left[1 + a_{SCP}(q-1)(2-q)^{\frac{1-q}{q-2}} \times 10^{2m_{min}}\right]^{\frac{2-q}{1-q}} - \left[1 + a_{SCP}(q-1)(2-q)^{\frac{1-q}{q-2}} \times 10^{2m_{max}}\right]^{\frac{2-q}{1-q}}} \quad ; m_{min} < m < m_{max}, \quad (12)$$

This doubly truncated magnitude distribution can be termed a bounded SCP recurrence law.

The appropriateness of this relationship can be evaluated by its compliance with regional data. This issue is later examined in

the practical example.

For our later PSHA equations, we will convert the continuous distribution of magnitudes into a discrete set of magnitudes. Probabilities of occurrence of these discrete sets of magnitudes, assuming that they are the only possible magnitudes, are computed as follows

$$P(M = m_j) \cong F_M(m_{j+1}) - F_M(m_j), \quad (13)$$

where $m_j$ is the discrete set of magnitudes, ordered so that $m_j < m_{j+1}$. This calculation assigns the probabilities associated

with all magnitudes between $m_j$ and $m_{j+1}$ to the discrete value $m_j$. As long as the discrete magnitudes are closely spaced, the approximation will not affect numerical results. In practice, magnitude spacing of 0.1 or less is appropriate.

Now, by substituting equation 12 instead of equation 2 in the classical PSHA (i.e., Equation 1), we present a non-extensive entropy-based approach to PSHA. We call the new approach non-extensive probabilistic seismic hazard analysis (NEPSHA). In this way, the physics-based recurrence law of the non-extensive SCP method will be entered into the hazard calculations.

As mentioned before, if the bounded SCP recurrence law shows a better match with regional data, the use of NEPHSHA will be on a more correct basis than the classical PSHA. It therefore may lead to more correct results of regional hazard. Thus, this approach provides a new possibility for modeling regional seismic conditions and hazard calculation.

As mentioned in the previous section, some modifications have been suggested for the SCP model (Silva et al., 2006, Telesca, 2012, Vallianatos et al., 2016, da Silva and Corso, 2021). Although these modifications are very helpful in

improving the method, the purpose of this paper is to provide a framework for incorporating the non-extansive models into the seismic hazard analysis process. Therefore, the basic approach of the SCP method is used as the basis of the work in this article. Obviously, by providing such a framework, it will be also possible to use modified SCP methods.

## 4 Application example

To highlight the effect of the proposed method on the hazard results, we implement the proposed method as a case study in

Tehran metropolitan. This city is located in one of the most active zones in the south of the Alborz seismic zone (Berberian and Yeats, 1999). For simplicity, in this study, the hazard of a single site from a single seismic fault was considered. Therefore, only one of the major active faults near Tehran, i.e., the North Tehran fault, was considered. Figure 1 shows the



location of the North Tehran fault seismic source. The selected site for PSHA was located at latitude and longitude coordinates of [35.59° N, 51.41° E].

In order to have a reliable estimate of the seismicity parameters, a homogeneous and complete earthquake catalog is required. In this study, the data were elicited from the USGS catalog (USGS, 2022), that covers the earthquake events from the fourth century BC to 2022. However, since, there is no clear approach to include historical earthquakes in the estimation of seismicity parameters using the SCP method; it was decided to neglect the historical earthquakes in this study. So only instrumental earthquakes (i.e., those earthquakes recorded after 1900AD) are considered here. After unifying magnitude

units using the Mousavi-Bafrouei et al. (2014) relationships, the dependent shocks have been removed from the earthquake catalog, using the time and distance windows methods proposed by Gardner and Knopof (1974) and Uhrhammer (1986).

The computer program SEISRISK II was used to estimate the GR seismicity parameters. Also, the SCP seismicity parameters have been calculated using a code written in the R language (R Core Team, 2021) based on the maximum likelihood method (Telesca, 2012). In this study, the parameters of both GR and SCP methods have been calculated based on

the same data and assumptions. Table 1 demonstrates the seismicity parameters of the GR and SCP methods. Figure 2 shows the fitted curves of GR and SCP. In this figure, the Empirical Cumulative Distribution Function (ECDF) of Observed data is also shown. Obviously, the SCP has a better fit for the data.

Other required information for seismic hazard analysis, including the fault geometry and location; the earthquake magnitude limits in the given region, $M_{min}$, and $M_{max}$; ground motion prediction equations, and local site characteristics, were considered

identical in both PSHA and NEPSHA and extracted from eligible articles (Nicknam et al., 2014; Yazdani et al., 2017).

Figure 3 shows the results of PSHA and NEPSHA for the selected site in the Tehran metropolitan area in terms of hazard curves for the selected site in the Tehran. As shown in Figure 3, the annual probability of exceedance (APE) is identical for both approaches (PGA = 0.01g). As the PGA increases, the difference between the two approaches hazard also increases. The APE obtained from the NEPSHA is greater than the value obtained from the PSHA. For PGAs greater than 0.1g, the

difference is approximately constant. Therefore, it can be concluded that the NEPSHA approach gives higher results, especially in higher PGAs. Also, the uniform hazard spectra (UHS) with 5% damping, based on the classic PSHA and NEPSHA with a probability of exceedance of 10% and 2% in 50 years, are shown in Figures 4 and 5, respectively. In the hazard spectrum curves for 2% exceedance probability in 50 years (figure 4), the values obtained based on NEPHSA are higher than those obtained from classic PHSA. The difference is considerable in the period range of 0.2 to 1 s, corresponded

to the height range of usual urban buildings. The difference gets smaller for tall buildings. In the uniform hazard spectra for 10% exceedance in 50 years (figure 5), the same behavior is observed, but in the high periods, the two curves are closer to each other than in the previous case.



## 5 Conclusion

Magnitude-frequency or recurrence relationship is an essential component of PSHA, which provides the cumulative rate of
occurrence of earthquakes within a seismic source zone as a function of magnitude. For many years, the Gutenberg-Richter relationship has been the governing paradigm in the energy distribution of earthquakes. However, the Gutenberg-Richter relationship still fits well with medium-sized earthquakes, but in small and large magnitude earthquakes, it deviates significantly. The core idea of this paper is that replacing the statistics-based equation of Gutenberg Richter's with an equation based on the physics of events can improve the hazard results. Here, the model presented by Sotolongo Costa and
Pasada (2000) on the interaction of barrier and asperity (SCP model) was developed and included in the PSHA process. The irregular geometry of the interacting plates and the fragments filling the space between them is the main factor considered in the numerical modeling of the SCP model. To this end, first, we derived the bounded SCP recurrence law. Then, By fitting this curve to the regional seismicity data, regional seismicity parameters are extracted. The better fit of this curve can be measured compared to Gutenberg Richter law. We founded the NEPSHA approach by rewriting the PSHA equation with a
bounded SCP recurrence law. The numerical example in the Tehran region shows the significant increase in the hazard of NEPSHA compared to PSHA. The difference is more considerable in the range of ordinary building height.

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






Table 1: Calculated seismicity parameters for the study area for GR and SCP

| GR | | SCP | |
|---|---|---|---|
| $a_{GR}$ | *b-value* | $a_{SCP}$ | *q-value* |
| 4.2744 | 1.26 | 5.71399e-09 | 1.670179 |




Figure 1: The location of the North Tehran fault, the border of the city, and selected sites for seismic-hazard analysis



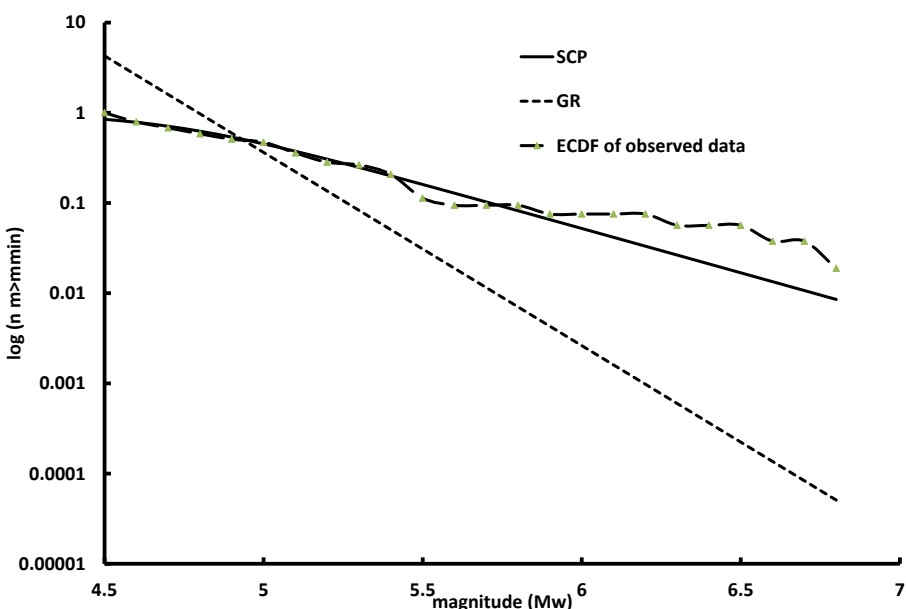


**Figure 2: Comparison of ECDF observed data with the GR and SCP models**

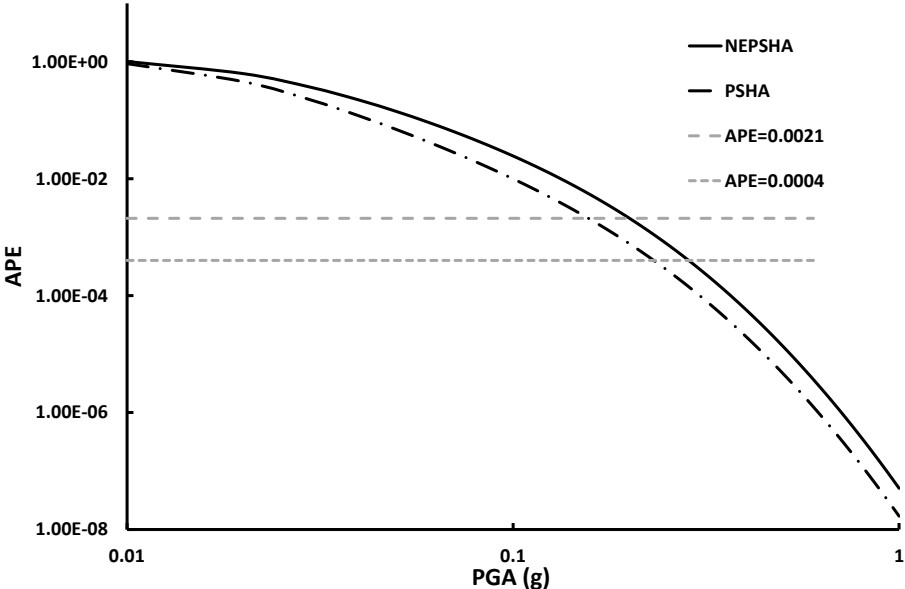

**Figure 3: Hazard curves based on PSHA and NE-PSHA approaches for PGA**




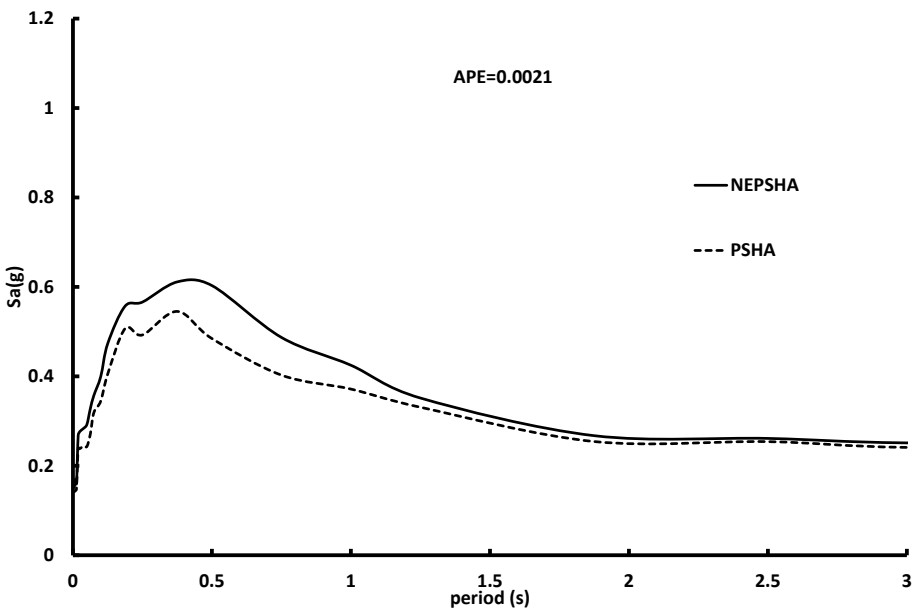

**Figure 4: The uniform hazard spectra based on NEPSHA and PSHA for the probability of exceedance 10% in 50 years**

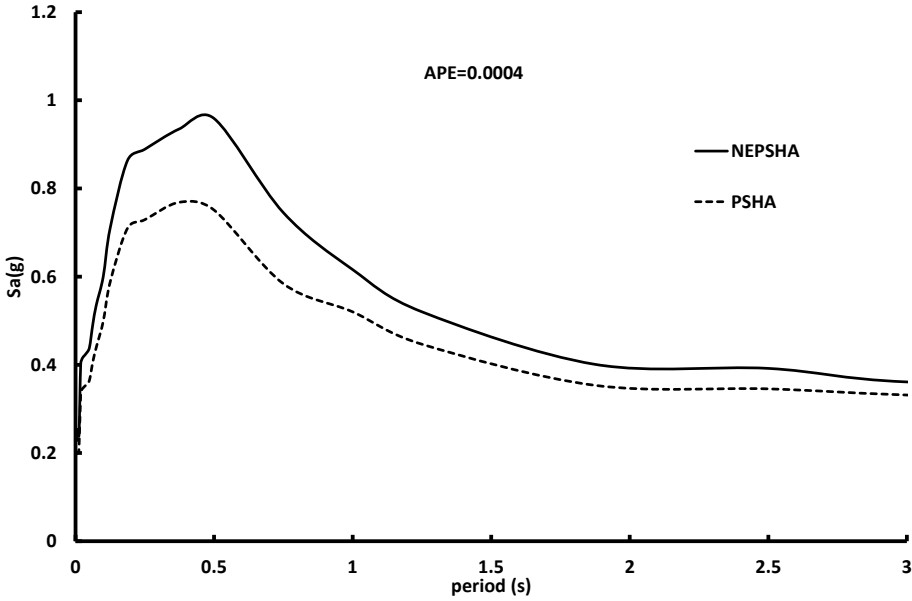

**Figure 5: The uniform hazard spectra based on NEPSHA and PSHA approaches for the probability of exceedance 2% in 50 years**