# Peer review of "A non-extensive approach to probabilistic seismic hazard analysis"

_Natural Hazards and Earth System Sciences, 2022_

## Author Comment (AC1)

**Manuscript NHESS-2022-214**
**Response to the respected reviewer**

Dear professor Vallianatos

Thank you for giving us the opportunity to submit a revised draft of the manuscript "A non-extensive approach to probabilistic seismic hazard analysis" for publication in the Natural Hazards and Earth System Sciences. We appreciate the time and effort that you and the respected reviewer dedicated to providing feedback on our manuscript and are grateful for the insightful comments on and valuable improvements to our paper. We have incorporated most of the suggestions made by the reviewer. Those changes are highlighted within the manuscript. Please see below, in blue, for a point-by-point response to the reviewer' comments and concerns.

**Reviewer' Comments to the Authors:**

Moatghed et al. in their paper "A non-extensive approach to probabilistic seismic hazard analysis" present a new approach for the probabilistic seismic hazard analysis (PSHA), in which they use the fragment-asperity model of Sotolongo-Costa and Posadas (SCP) to describe the frequency-magnitude distribution of earthquakes, instead of the well-known Gutenberg-Richter (GR) scaling law. Various studies during the last two decades have demonstrated that the SCP model, based on the concept of Tsallis entropy, it provides a physical model for the energy distribution of earthquakes. In addition, it provides in various cases a better fit to the observed frequency-magnitude distribution over a wider range of magnitudes compared to the GR law. Nonetheless, the well-known b-value can be deduced as a particular case in the SCP model. In this framework, the generalization of the classic PSHA by using the SCP model may provide better results regarding the estimation of seismic hazard. The paper presented by Moatghed et al. aims to contribute to this field and clearly falls within the scope of Natural Hazards and Earth System Sciences. The paper is generally well written and structured, but it needs some revisions before it can be further considered for publication. Some points that require further clarification are listed below. The main issue concerns the application of PSHA in the Tehran region in Section 4.

**Authors' reply:** Thank you for reaching out and providing us with valuable feedback. We found your comments extremely helpful and have revised accordingly.

1. The spatial distribution of earthquakes used in the analysis should be shown in a Figure, perhaps Fig.1.

**Authors' reply:** Fig. 1 has been modified by adding the suggested content.

2. The authors use earthquakes since 1900AD. Which is the magnitude of completeness of the catalogue during this period?

**Authors' reply:** The year 1900 is the beginning of the instrumental recording of earthquakes, and for this reason, it has been of fundamental importance in the past researches of the seismicity of the Tehran region. Based on the observations, the first event in this period was recorded in 1930, which definitely indicates incomplete data recording. For this purpose, the Kijko method that provided some considerations to solve this problem, are also included in these calculations. However, since the purpose of this article is to present the methodology, local issues have not been described much in order to summarize.
It is reminded that the advantage of the SCP relationship is in better matching with the range of low magnitudes (which GR relation probably does not show a good compatibility with them due to its

incompleteness) and high magnitudes (which probably does not have an accurate recording due to the saturation of the instrument), which in this example also shown this problem.

3. Present a Figure showing the cumulative number of earthquakes used in the analysis and the cumulative number after declustering to show its effectiveness.

**Authors' reply:** Thank you for your creative thinking. Fig. 2 has been modified by adding the ECDF of the decluttered data.

4. Which method was used to estimate the GR parameters? Obviously, in Fig.2 the GR law is not well implemented.

**Authors' reply:** Thank you for your consideration and accuracy. The GR parameters have been estimated based on the Kijko's maximum likelihood method. So, the following sentences are added to the main text (lines 168 and 169)

"The GR seismicity parameters (i.e., the rate of seismicity and b-value) are computed using the Kijko's maximum likelihood method (Kijko and Sellevoll 1989;Kijko, 2004). For this end, a MATLAB program (HA3) written by Kijko et al. (2016) has been utilized."

Also, you carefully point out the Incompatibility of GR parameters to observed data in Fig. 2. You are absolutely right. This is because we have mistakenly reported the value of a instead of b in this figure the α and β values $\alpha = a_{GR} \times ln(10)$ and $\beta = b\text{-}value \times ln(10)$ values instead of $a_{GR}$ and $b\text{-}value$ in figure 2 (and also in the Table 1). This mistake has led to the incorrect drawing of the GR curve. Accordingly, this figure was modified.

5. Provide confidence intervals for the parameter values in Table 1.

**Authors' reply:** We have added the suggested content to the manuscript on Table 1.

6. Revise all calculations of PSHA based on the better estimation of the GR parameters. Show in Fig.3-5 the revised calculations and the corresponding confidence intervals.

**Authors' reply:** Thank you for your accuracy. Based on your comment, we revised both PSHA and NEPSHA based on modified parameters. It should be noted that in the revised analyses, a better local attenuation relationship, i.e., Yazdani and Kowsari (2013) is used (instead of Ramazi and Schenk (1994)). Accordingly, the results of probabilistic seismic hazard analyses (in figures 3 and 4) were updated.

7. Provide more information on how the uniform hazard spectra are calculated.

**Authors' reply:** Based on your comment, the following sentences are added to the main text (lines 179-181):

"These spectra are essentially derived from hazard curves, and cover a broad range of spectral periods. To construct UHS from a set of hazard curves, one can conceptualize this process as simply extracting from multiple hazard curves all of the intensity measure levels for a given APE."

8. Some minor comments concern:
   Correct to "Posadas" in Line 185.
   Correct to "NEPSHA" in Line 135.
   Refer to other relevant studies that use the Tsallis entropy approach to identify precursors in the earthquake generation process, such as Vallianatos et al. (2014), Physica A.
   Refer to other relevant studies that review the non-extensive approach in earthquakes and tectonics, such as Vallianatos et al. (2016), Proc. R. Soc. A.

**Authors' reply:** Thank you for your kind interest. The corrections were done and the mentioned references were cited in the paper.

---

## Author Comment (AC3)

**Manuscript NHESS-2022-214**
**Response to the respected reviewer #2**

Dear professor Telesca

From the outset we would like to convey our appreciation for the thorough, critical and fair review of our manuscript. You raise several important points and we believe that we can address all of them in a satisfactory manner. Moreover, we can identify that in doing so that our manuscript will have considerably been improved, which we appreciate also greatly.

Here is a point-by-point response to your comments and concerns.

**Reviewer' Comments to the Authors:**
The paper proposes to change the classical frequency-magnitude distribution (the GR scaling law) in the PSHA with the non-extensive frequency-magnitude distribution derived from the SCP model. The aims of the paper fall within the scope of the journal. The study is clearly written and structured. However, some points need to be clarified.
**Authors' reply:** Thank you for reaching out and providing us with valuable feedback. We found your comments extremely helpful and have revised accordingly.

1.  Page 5, the authors say "In this equation, unlike the non-extensive expression of Telesca (Telesca, 2012) in which the catalog completeness magnitude is used, we include the minimum earthquake magnitude of engineering significance". How the "minimum earthquake magnitude of engineering significance" is defined? If the completeness magnitude is not a fundamental parameter, why the authors at page 7 say "In order to have a reliable estimate of the seismicity parameters, a homogeneous and "complete" earthquake catalog is required."

**Authors' reply:** The completeness magnitude is a key factor in estimating the seismicity parameters. In this statement, we do not mean that this parameter is less important. This merely corresponds to the fact that in the evaluation of PSHA integral (not in seismicity analysis), mainly the minimum earthquake magnitude of engineering significance is used. This parameter is defined as "the smallest magnitude of earthquake that is capable of generating potentially damaging levels of ground shaking" (Bommer and Crowly, 2017). However, we have modified our statement in this section to avoid misleading information.

Bommer, J. J., & Crowley, H. (2017). The purpose and definition of the minimum magnitude limit in PSHA calculations. *Seismological Research Letters*, *88*(4), 1097-1106.

2.  The authors just say that the GR parameters were calculated by using the SEISRISK II software. However, Fig. 2 shows that the GR law does not fit at all the ECDF, which can be easily fitted by a straight line whose slope gives the estimate of the b-value that should be smaller than that indicated in Table 1.

**Authors' reply:** We thank you for pointing out this problem. You are absolutely right. This is because we have mistakenly reported the α and β values (i.e., $\alpha = a_{GR} \times ln(10)$ and $\beta = b\text{-}value \times ln(10)$) instead of $a_{GR}$ and *b-value* in this figure (and also in the Table 1). This mistake has led to the incorrect drawing of the GR curve. Accordingly, this figure was modified.

3. The epicentral distribution of the earthquakes needs to be shown.

**Authors' reply:** Based on your comment, the epicentral distribution of the earthquakes has been added to figure 1.

4. The declustering is performed by using the Gardner and Knopoff method with Uhrhammer window. It is known that this method can also be used with the Grunthal window (van Stiphout et al., 2012, doi:10.5078/corssa-52382934. http://www.corssa.org). Why did the authors use Uhrhammer?

**Authors' reply:** In this example, we aim to illustrate the difference between PSHA and NEPSHA results. The use of Uhrhammer window here for declustering does not mean that it is superior to other methods. We used this window because it is a known method and has been used in many seismicity studies.

5. Recently, Mizrahi et al. (Seismol. Res. Lett. 92, 2333–2342, 2021 doi: 10.1785/0220200231) concluded that "declustering should be considered as a potential source of bias in seismicity and hazard studies", since the GR parameters depend on the method of declustering. Thus, I think, the paper would be improved if the authors discuss and compare the results obtained after applying also another method of declustering besides that cited in the paper.

**Authors' reply:** Thank you for your comment and suggestion. Generally, the main purpose in this work is to develop an efficient scheme to PSHA based on the fragment-asperity (SCP) model instead of the Gutenberg-Richter (GR) scaling law. In this paper, the computational framework of the proposed NEPSHA method is presented. We provide here a simple example only to describe and evaluate the proposed framework. Obviously, in practical applications, there are some issues that can be investigated and evaluated (e.g., effect of ground motion prediction equation selection on PSHA results, effect of the catalog selection on seismicity parameters, and sensitivity of PSHA results to the declustering methods). In this context, the valuable results provided by Mizrahi et al. can be discussed and investigated. But we believe that detailed dealing with these issues can overshadow the main purpose. Undoubtedly, the detailed examination of these cases can be a research topic itself. But we are worried that addressing them in this manuscript will mislead the readers. Finally, if the honorable referee considers it necessary, we are ready to add it to the present work.

---

## Author Response (AR2)

Dear professor Filippos Vallianatos

We would like to thank you and reviewers for careful and thorough reading of this manuscript and for the thoughtful comments, which help to improve the quality of this manuscript. Below, we also provide a point-by-point response on how we have addressed each of the reviewer's comments:

Response to Anonymous referee #1

In the revised version of the paper "A non-extensive approach to probabilistic seismic hazard analysis" the authors have addressed most of the issues related to the previous version of the manuscript. Therefore, I recommend its publication after some minor corrections listed below.

***Thank you for these positive comments. Below is our response to the issues raised in the review.***

1) The caption of Fig.1 should describe all aspects presented, as for instance what the symbols show (earthquakes).

***Response: Corrected as suggested by the reviewer.***

2) In Table 1 there is no point in showing the parameter values up to six decimal digits. Round the values up to 2 or 3 decimal digits. Add the decimal operator to the upper confidence value of q.

***Response: Thanks for your accuracy. The correction has been made.***

3) After Equation 1, rewrite the parameter v to be similar to the one shown in the equation.

***Response: The correction has been made.***

4) In the paragraph before section 4 in page 6, correct "extansive" to "extensive".

***Response: The correction has been made.***